# Prolonged Social Isolation, Started Early in Life, Impairs Cognitive Abilities in Rats Depending on Sex

**DOI:** 10.3390/brainsci10110799

**Published:** 2020-10-30

**Authors:** Nataliya A. Krupina, Sophie D. Shirenova, Nadezhda N. Khlebnikova

**Affiliations:** Laboratory of General Pathology of the Nervous System, The Institute of General Pathology and Pathophysiology, 125315 Moscow, Russia; shirenova.jr@gmail.com (S.D.S.); nanikh@yandex.ru (N.N.K.)

**Keywords:** prolonged social isolation, male and female Wistar rats, open field, habituation, Morris water maze, spatial memory, passive avoidance paradigm, corticosterone, adrenal glands, spleen, thymus

## Abstract

Background: The chronic stress of social isolation is a valid predictor of cognitive pathology. This study aimed to compare the effects of long-term social isolation on female versus male Wistar rats’ learning and memory. We hypothesized that prolonged social isolation stress, which starts early in life, would affect learning in a sex-dependent manner. Methods: Social isolation started at the edge of early to mid-adolescence and lasted 9 months. The rat’s cognitive abilities were assessed by habituation and reactivity to novelty in the open field (OF) test, spatial memory in the Morris water maze (MWM), and the conditioned passive avoidance (PA) reflex. Basal serum corticosterone levels were assessed using an enzyme-linked immunosorbent assay. Results: Regardless of the housing conditions, females habituated to the OF under low illumination slower than males. Under bright light, the single-housed rats showed hyporeactivity to novelty. In the MWM, all the rats learned to locate the platform; however, on the first training day, the single-housed females’ speed was lower relative to other groups. Four months later, in the post-reminder probe trial, the single-housed rats reached the area around the platform site later, and only males, regardless of housing conditions, preferred the target quadrant. Single-housed rats, irrespective of sex, showed a PA deficit. There was a more pronounced conditioned fear in the single-housed males than in females. In both male and female rats, basal corticosterone levels in rat blood serum after 9 months of social isolation did not differ from that in the group-housed rats of the corresponding sex. Meanwhile, females’ basal corticosterone level was higher than in males, regardless of the housing conditions. The relative weight of the adrenal glands was increased only in single-housed females. Conclusions: Under long-term social isolation, started early in life, single-housed females compared with males showed more pronounced cognitive impairments in the MWM and PA paradigm, findings that specify their greater vulnerability to the stress of prolonged social isolation.

## 1. Introduction

Chronic or recurring stress increases the risk of central nervous system (CNS) impairments and often leads to psychoneurological disorders through several synaptic plasticity mechanisms and changes the activity of stress-related systems, in particular, the hypothalamic–pituitary–adrenal (HPA) axis [1,2]. HPA axis dysfunction under chronic stress impairs the formation of systems involved in learning, memory, and emotions in a developing brain [3,4]; the outcome is sex-dependent [5]. 

Social isolation/deprivation is a form of chronic stress [6,7]. In humans, social isolation is a valid predictor of cognitive pathology [8] and is associated with cognitive decline [9,10]. Early-life stress induced by social and sensory deprivation gives rise to disruptions in sensory modulation [11] and social and emotional regulation. Furthermore, early-life social isolation stress affects brain development and neuroplasticity, which may lead to behavioral alterations and motor skill deficits [12,13,14,15,16,17,18]. Later in life, a behavioral phenotype develops, which is characterized by a decline in cognitive abilities: atypical associative learning, spatial memory deficit, impairments of executive functions (inhibitory control, cognitive flexibility, and sustained attention) with a deficit in affective functioning (emotion regulation) [17,19,20]. Such changes increase the risk of social maladjustment and psychopathology.

Sex differences have been found in brain reorganization and formation of neuroendocrine pathways during puberty [21,22,23], which account for differences in glucocorticoid system reactivity to psychosocial stress in males versus females. Girls compared with boys and female compared with male rodents show a more robust stress reaction to physical and psychological stimuli as judged by an increase in glucocorticoid secretion [24]; for a review, see [25]. Short- and long-term changes in behavioral stress response and HPA axis activity under chronic stress (including social isolation stress) in puberty depend on gender (for a review, see [26]). In summary, judging by behavior and the HPA axis state, females are more sensitive to the social isolation stress than males [27,28]. However, some contradicting data have also been reported [29]. 

Few experimental studies have compared the effects of social isolation on cognitive abilities in male and female rodents in the same experiment, and the results are controversial. In C57BL/6J mice, 1-week rearing in isolation facilitated contextual fear memory (freezing in classical conditioned defense reflex) in females but not males, while it affected neither fear extinction nor fear memory recall in both sexes [30]. After 8 weeks of social rearing, adult C57BL/6J mice showed no between-sex differences in spatial memory assessed with the Morris water maze (MWM): Both males and females demonstrated decreased memory acquisition, memory consolidation, and retrieval [31]. However, in the same study, the authors reported that impairments in episodic-like memory revealed in both sexes in the novel object recognition test were more profound in females. In male and female Sprague Dawley rats, 21-week social isolation starting from postnatal day 21 (PND21) decreased contextual fear memory in the conditioned defense reflex, although, only in males, there was an increase in basal adrenocorticotropic hormone (ACTH) but not corticosterone (CORT) in blood plasma [32]. The stress of the startle-response procedure enhanced the release of both ACTH and CORT in isolated male rats relative to control males. In Long–Evans rats weaned from the dams on PND23, 5–6 weeks of social isolation impaired habituation (non-associative learning) in both males and females [33]. Another study reported that 7–9 weeks of social isolation impaired social preference in females but not in males [34].

Comparative studies that aim to find sex differences in the effects of chronic social isolation on cognitive functions in rodents can benefit from additional assessments of organs, the weight of which may indirectly change under stress and indicate alterations in the HPA axis that characterize so-called “isolation syndrome” (endocrine organs—adrenal and pituitary glands; lymphatic organs—thymus and spleen) [35,36]. The direction of such changes typically depends on the sex of the animals, but it may also depend on, for example, the strain or duration of social isolation [37,38,39,40].

In our previous studies, we have revealed the deficit of short-term acoustic startle habituation and locomotor habituation in the open field (OF) test in male Wistar rats after 2 and 3 months of social isolation [41,42]. However, there was no spatial memory impairment after 3 months of isolation. Considering the above, we hypothesize that prolonged social isolation stress, which starts early in life, would affect learning in a sex-dependent manner. In the current study, we compared the effects of long-term post-weaning social isolation on learning and memory in female versus male Wistar rats. We extended social isolation up to 9 months. We assessed cognitive abilities by short-term habituation and reactivity to novelty in the OF test, spatial memory in the MWM (the dynamics of short- and long-term spatial memory), and the conditioned passive avoidance (PA) paradigm. We measured basal CORT levels in blood serum collected after nine months of social isolation. Additionally, we evaluated the relative weights of the adrenal glands, thymus, and spleen.

## 2. Materials and Methods

### 2.1. Animals

We obtained male and female Wistar rats (*n* = 69) from the nursery of the Institute of General Pathology and Pathophysiology. The animals were kept under standard vivarium conditions at natural light/dark cycle with free access to water and food (dry, balanced fodder produced by Laboratorkorm OOO, Moscow, Russia).

We performed all manipulations on animals in accordance with EU Directive 2010/63/EU and the Rules of Good Laboratory Practice in the Russian Federation approved by Order N 199h (1 April 2016) of the Ministry of Health Care, under supervision of the Ethics Committee of the Institute of General Pathology and Pathophysiology (project approval protocol No 6 of 23.11.2018; the final approval protocol No 3 of 16.06.2020).

We considered the day of birth as PND0. On PND1, dams were left with five males or five females, each from a different litter. This regrouping minimized the effects of genotype and accounted for different maternal care. Rats were weaned from the dams on PND 26. At 1 month of age (PND33 for females and PND37 for males), we assigned rats to one of four groups: group-housed females (*n* = 17), single-housed females (*n* = 16), group-housed males (*n* = 20), and single-housed males (*n* = 16). Group-housed rats continued being kept in the groups formed initially after separation from mothers (4–5 rats per 37.0 × 57.0 × 19.0 cm cage). Single-housed rats were kept in a 36.5 × 20.5 × 14.0 cm cage.

### 2.2. Behavioral Testing

At 1, 3, 5, and 8.5 months of age, rats from all the groups performed two OF tests in the automated open field (aOF) and classic open field (cOF) (see Section 2.3). At 6 and 9.5 months of age, we trained the rats in the MWM to assess the spatial memory (see Section 2.4), and at the age of 10 month, we trained the rats to perform the PA task (see Section 2.5). Upon completion of behavioral testing, we sacrificed the animals. Figure 1 presents the schedule of behavioral testing, with the PND for each procedure.

### 2.3. Open Field Test

We assessed locomotor activity and exploratory behavior using the aOF and the cOF at four times: in 1-month-old rats (initial assessment before group assignation), and in 3-, 5-, and 8.5-month-old rats (after 2, 4, and 7.5 months of appropriate housing, respectively). The aOF utilized a 48 × 48 × 21 cm arena with transparent walls was used (Opto-Varimex, Columbus Instruments, Columbus, OH, USA). We monitored the behavior under mild room light (17 lx) with activity measured by the number of photo-beam breaks over 10 min. The cOF utilized a larger white-colored wooden circular arena (120 cm in diameter with an outer wall height of 28 cm) divided into 20 cm squares. In the first three minutes, the behavior was recorded under bright light (six 60 W lamps, arranged in a circle with a diameter of 60 cm suspended 80 cm above the center, 500–510 lx in the center, 400–410 lx in the peripheral area near the wall), while in the fourth minute of the session, the behavior was recorded under soft red light (a 40 W red lamp suspended 80 cm above the center). We assessed locomotor activity by traveled distance (cm in the aOF, and the number of squares under visual control in the cOF), and we measured exploratory behavior as the number of vertical rearing postures. Rats were not habituated to the activity chambers prior to the OF tests. 

The assessment of the reactivity to novelty in rats when testing in the OF is based on a well-known increase in the locomotor activity in rats under red light compared with white light [43]. Sudden change in lighting is a novelty factor. We evaluated the reactivity to novelty as follows: 

(the distance traveled in the first minute under a bright light)/(the distance traveled in the fourth minute of a session under the red light). 

We assessed habituation (non-associative learning) in the aOF as a decrease in the distance traveled and the number of vertical rearing postures during the session. We cleaned the aOF and cOF arenas with ethanol before each animal was tested.

### 2.4. Morris Water Maze

We used the MWM to test spatial learning and memory in animals. Briefly, the MWM consisted of a black circular tank (160 cm in diameter, outer border height of 75 cm, inner border height of 60 cm) filled with water (25 ± 1 °C) to a depth of 30 cm (the apparatus is a part of the automated video tracking system VideoMot2, TSE System, Bad Homburg vor der Höhe, Germany). The software virtually divided the maze into four equal quadrants. We placed a transparent escape platform (14 cm in diameter) at a fixed position (in the target quadrant) 1.5 cm below the water surface. The software also virtually marked the platform site, the area around it (10 cm from the rim of the platform), and the border area (outer 20% of the tank radius). We placed visual keys (objects and images) outside the tank.

The training that started in 5.5-month-old rats was similar to the procedure used by Chakravarthi and Avadhani [44], with additional recommendations made by other studies [45]. Briefly, a rat performed four acquisition trials to find the hidden platform on each of the four training days. With four principle start points, we presented each rat with all the starting locations in the same pseudorandom order in a day. The maximum trial duration was 2 min. If a rat found the platform in under 2 min, it was allowed to stay on the platform for 15 s. If the rat did not find the platform within 2 min, it was manually directed to the platform and left on it for 15 s. After each trial, the rat was dried with a towel and placed in a cage with sawdust. Inter-trial intervals were at least 5 min. On day 5, 24 h after the last training trial, we conducted a probe trial (Probe 1). We removed the platform from the tank and placed a rat in the quadrant opposite to the target one. The probe trial was 2 min. Four months later, when rats were 9.5-month-old, they performed another probe trial (Probe 2). The next day, 24 h after Probe 2, rats received two Reminder trials with a 5-min interval between them. For the Reminder trials, we returned the platform to the tank. The starting positions were in the two quadrants adjacent to the target one. Each Reminder trial lasted a maximum of 2 min. The following day, 24 h after the last Reminder trial, we conducted Probe 3. 

We analyzed six indexes for the acquisition, reminder, and probe trials: distance traveled, average speed, latency to reach the platform site, latency to reach the area around the platform site, relative time spent in the target quadrant, and relative time spent in the border area. We averaged the data from the training days and the day of Reminder trials between the trials of the day.

### 2.5. Passive Avoidance Paradigm

At the age of 10 months, after 9 months of appropriate housing, we subjected the rats to the PA task. One single-housed male lost more than 20% of body mass relative to mean body mass of the single-housed males, and we excluded it from this experiment. We placed a grid floor arena with dark and light compartments inside a sound-proof chamber (Multi Conditioning System, TSE, Bad Homburg, Germany). A guillotine door separated the compartments. We conducted experimental procedures at 50 dB white noise. Rats first underwent a habituation period during which we placed them in the light compartment of the arena at 156 lux illumination and allowed them to freely explore it for 60 s. Then the guillotine door opened, and rats could enter the dark compartment. The maximum time to enter the dark compartment was 120 s. The latency to enter the dark compartment (sec) was defined as the time when the rat passed into the compartment with all four paws. One second after a rat crossed into the dark compartment, the door closed. We allowed a rat to explore the dark compartment for 30 s, after which we removed it from the arena and returned to their home cages. If the rat did not enter the dark compartment within 120 s, the habituation stage ended, and we excluded it from the experiment (three group-housed males, one single-housed male, and one single-housed female). The following day, 24 h later, we subjected rats to an acquisition trial. We again placed a rat in the light compartment facing away from the guillotine door. Five seconds later, the door opened, and we recorded the latency to enter the dark compartment (up to a maximum of 120 sec). One second after the transfer, the door closed, and after 5 s, an electrical foot shock was delivered through the grid floor (AC, 0.5 mA, 1 Hz) for 5 s. Thirty seconds later, we removed the rat from the arena and returned it to its home cage. Three males (one group-housed and two single-housed) and one single-housed female failed to produce a clear motor response to the electrical foot shock, and therefore, we excluded them from further analysis. The retention trials occurred 24 h following the acquisition trial (Retention 1) and 7 days after Retention 1 (Retention 2). The procedure was similar to the acquisition trial except this time rats did not receive the electrical foot shock, and we recorded the latency to enter the dark compartment (up to 300 s). When the animal had not entered the dark compartment by the end of the trial, it received the maximum value (300 s). We assessed the latency to enter the dark compartment and avoidance levels (the proportion of animals that avoided possible electrical foot shock in retention trials, relative to the number of animals in the group).

In addition, on each Retention trial, we counted the number of peeps when the animal put its head into the dark compartment and the number of incomplete transfers when the rat entered the dark compartment with 1–3 paws. We also assessed the quality of the conditioned PA reflex by the method described by Pavlova et al. [46], using a point system: 5, the rat does not enter and does not peep into the dark compartment; 4, the rat does not enter, but peeps into the dark compartment; 3, the rat does not enter, but peeps into and makes an incomplete transfer; 2, the rat enters the dark compartment with a higher latency than in the control experiments; and 1, the rat enters the dark compartment with latency not higher than in the control experiments.

### 2.6. Enzyme-Linked Immunosorbent Assay

Upon completion of behavioral testing at 10 months of age, before decapitation and sampling, we transferred the rats from the vivarium to the laboratory room preserving the group-housed animals in the same groups and the single-housed rats–separately. After the transfer, the animals were allowed to recover for about an hour. We assessed the basal serum CORT level and did not subject the rats to any additional stress before decapitation and sampling. We decapitated the animals using a guillotine. We isolated the brain structures and froze them in liquid nitrogen for subsequent molecular analysis (data for BDNF, synaptophysin, glucocorticoid receptor, and some other proteins’ expression in the frontal cortex, hippocampus, hypothalamus, and striatum under process). We collected trunk blood samples immediately after decapitation, at the same time of day for male and female rats, ante meridiem. We used special tubes with a coagulation activator and gel (VACUETTE^®^ Blood Collection Tubes, Greiner-Bio-One GmbH) to collect blood samples. We centrifuged the samples to obtain serum, and froze them at −83 °C for further CORT assessment. We purchased rat CORT enzyme-linked immunosorbent assay (ELISA) kits from DRG Instruments GmbH (Marburg, Germany). We used a separate frozen sample aliquot from each subject for each assay, and we performed the assays in duplicate according to the manufacturer’s recommendations. We read the plates on an Immunochem-2100 plate reader (High Technology, Inc., Walpole, MA, USA) and then plotted a four-parameter logistic curve using Myassays.com to extrapolate data. Serum CORT levels are expressed in ng/mL.

### 2.7. Statistical Analysis

We analyzed data using Statistica 8.0 software (StatSoft Inc., Tulsa, OK, USA). We checked all variables for a normal distribution using the Kolmogorov–Smirnov test. We analyzed data using multivariate repeated measures analysis of variance (ANOVA) for sex (two levels: male and female) and housing (two levels: group housed and single housed). The repeated measures were the following: for the aOF, cOF tests, and animal weight, age (four levels: 1, 3, 5, and 8.5 months); for the aOF test, time (10 levels: minutes 1 to 10); for the PA paradigm, stage (four levels: Habituation, Acquisition, Retention 1, and Retention 2) followed by Newman–Keuls post hoc test. We analyzed data from the MWM, serum CORT concentration, stress-related organ weights, and animal weight using two-way ANOVA for sex and housing, followed by Newman–Keuls or Fisher’s least significant difference (LSD) post hoc test. In the MWM, we used a one-sample Student’s *t*-test to decide whether the preference for the target quadrant was random (compared to the 25% critical value). In the PA paradigm, we compared avoidance levels between groups using Fisher’s exact test (two-tailed). According to the results of the preliminary verification, we rejected the hypothesis of normal data distribution for the number of peeps and the number of incomplete transfers in PA as well as the quality of PA (in points); hence, we used a nonparametric unpaired Mann–Whitney U test for independent variables (two-sided) for comparative analysis with false discovery rate (FDR) control [47] to correct for multiple comparisons. Data are given as mean ± standard error of the mean (SEM). The accepted significance level was 5%.

## 3. Results

### 3.1. Open Field Test

#### 3.1.1. Automated Open Field Test

*Locomotor Activity.* Three-way repeated measures ANOVA revealed a significant sex main effect (*F*(1,63) = 25.244, *p* < 0.001): Females traveled longer distances over 10 min (Newman–Keuls test, *p* < 0.001). The analysis also revealed a significant age effect (*F*(3,189) = 10.051, *p* < 0.001) and age × sex interaction (*F*(3,189) = 13.650, *p* < 0.001) (Figure 2A). There was no age × housing interaction (*F*(3,189) = 0.716, *p* = 0.543). The sex × housing × age interaction almost reached statistical significance (*F*(3,189) = 2.419, *p* = 0.068). Females exhibited higher locomotor activity at every examined age compared with the age of 1 month (the difference is not shown in Figure 2A). In 8.5-month-old males, locomotor activity was lower than at the age of 3 months. Females were more active than males when compared at 3, 5, and 8.5 months of age, while 1-month-old males and females did not differ (Figure 2A). 

*Habituation assessed by locomotor activity*. In the 1-month-old rats, ANOVA revealed a significant time main effect (*F*(9,585) = 126.484, *p* < 0.001); all of the animals showed a gradual decrease in their locomotor activity. There was neither a time × sex nor a time × housing interaction. In the 3-month-old rats, there was a significant time main effect (*F*(9,567) = 54.552, *p* < 0.001) and a time × sex interaction (*F*(9,567) = 2.244, *p* < 0.018). Both males and females showed decreased locomotor activity with time, but females habituated slower than males (Figure 3A). The within-group post hoc analysis showed that in the 3-month-old females, the decrease in the locomotor activity stopped only at the eighth minute, and in the 3-month-old males, for the third minute. The motor activity in females exceeded that in males at the first, third, and fourth minutes. In the 5-month-old rats, there was a significant main effect of time (*F*(9,567) = 26.888, *p* < 0.001), but there was no time × sex interaction. The habituation rate did not differ between groups. In the 8.5-month-old rats, again both the time main effect (*F*(9,585) = 50.100, *p* < 0.001) and the time × sex interaction (*F*(9,585) = 2.747, *p* < 0.004) were significant. Females showed increased locomotor activity compared with males only in the first four minutes of the session (Figure 3B). According to the within-group analysis, the reduction in the locomotor activity decreased at the fifth minute for 8.5-month-old females and at the third minute for the 8.5-month-old males.

In the 3-month-old rats, ANOVA revealed a significant sex × housing interaction (*F*(1,63) = 4.443, *p* = 0.039). Group-housed females were more active relative to group-housed males (Newman–Keuls test, *p* < 0.001). Single-housed males and females did not differ.

*Exploratory behavior.* There was a significant sex × housing interaction (*F*(1,63) = 4.807, *p* = 0.032). The total number of rearing postures in 10 min was higher in the group-housed females compared with the group-housed males (Newman–Keuls test, *p* = 0.024), but there was no difference between single-housed males and females (Newman–Keuls test, *p* = 0.867). The sex and housing main effects were not significant. The analysis revealed a significant age main effect (*F*(3,189) = 48.642, *p* < 0.001). All the animals showed a lower number of rearing postures at every other stage compared with the age of 1 month (Figure 4A). There was a significant age × sex interaction (*F*(3,189) = 7.865, *p* < 0.001). In the 5-month-old rats, females showed a greater total number of rearing postures than males.

*Habituation assessed by exploratory behavior*. The time main effect was significant at every examined age (month 1: *F*(9,585) = 42.152, *p* < 0.001; month 3: *F*(9,567) = 35.560, *p* < 0.001; month 5: *F*(9,567) = 26.888, *p* < 0.001; and month 8,5: *F*(9,585) = 28.997, *p* < 0.001). All the animals showed habituation to the open field environment, as judged by a gradual decrease in the number of rearing postures throughout the testing session. Neither the time × sex nor the time × housing interaction was significant. However, in 8.5-month-old rats, the time × housing interaction almost reached the accepted significance level (*F*(9,585) = 1.883, *p* = 0.052), which would indicate delayed habituation in the single-housed rats relative to group-housed animals.

Similar to the locomotor activity, in the 3-month-old rats, ANOVA revealed a significant sex × housing interaction (*F*(1,63) = 6.640, *p* = 0.012). The total number of rearing postures was higher in the group-housed females compared with the group-housed males and single-housed females (Newman–Keuls test, *p* = 0.021, and *p* = 0.031, respectively).

#### 3.1.2. Classic Open Field Test

*Locomotor activity*. ANOVA yielded significant sex (*F*(1,64) = 23.230, *p* < 0.001), housing (*F*(1,64) = 4.337, *p* = 0.041), and age (*F*(3,192) = 38.012, *p* < 0.001) main effects. Females were more active than males (Newman–Keuls test, *p* < 0.001) (Figure 5A), and single-housed rats were more active than group-housed rats (Newman–Keuls test, *p* = 0.021) (Figure 5B). Locomotor activity decreased with age in both males and females. The sex × housing, age × housing, and sex × age × housing interactions were not significant. However, the age × sex interaction presented a trend for statistical significance (*F*(3,192) = 2.574, *p* = 0.055).

To compare correctly the effects in the two types of OF tests, we evaluated sex and housing effects with age on the total locomotor activity for the first 3 min of testing in the aOF test (Figure 1B). As in the cOF test, ANOVA revealed a significant sex effect (*F*(1,63) = 27.612, *p* < 0.001): Females were more active (Newman–Keuls test, *p* < 0.001). ANOVA also revealed a significant age main effect (*F*(3,189) = 3.198, *p* = 0.025): At 8.5 months old, the rats traveled shorter distances than at 3 months old. The age × sex interaction was significant (*F*(3,189) = 11.192, *p* < 0.001): In females, locomotor activity increased with age, but this measure decreased in males (Figure 1B). Besides, the sex × housing × age interaction was significant (*F*(3,189) = 2.847, *p* = 0.039). The Newman–Keuls post hoc test specified that only in the group-housed rats did locomotor activity change with age: increased in females and decreased in males. In the group-housed rats, females were more active at 3 and 5 months of age. In the single-housed rats, females were more active than males only at 8.5 months of age. 

*Exploratory behavior*. The analysis of rearing yielded only a significant age effect (*F*(3,189) = 5.718, *p* < 0.001). The rats made significantly fewer vertical rearing postures at 5 and 8.5 months than at 1 and 3 months of age (Newman–Keuls test, *p* ≤ 0.031 for all the comparisons). When estimated for the first three minutes in the aOF, there was a sex main effect on the total number of rearings (*F*(1,63) = 5.654, *p* = 0.020). In addition, the age × sex interaction was significant (*F*(3,189) = 7.858, *p* < 0.001) (Figure 4B). In females, the number of rearings was higher than in males at every age except for 1 month.

*Reactivity to novelty*. Isolated rats demonstrated a decrease in the reactivity to novelty (housing main effect: *F*(1,61) = 19.907, *p* < 0.001). The age effect and age × housing interaction were also significant (respectively: *F*(3,183) = 7.191, *p* < 0.001 and *F*(3,183) = 2.736, *p* = 0.045). In the group-housed but not in the single-housed rats, the reaction to novelty increased with age: At 5 and 8.5 months, it exceeded the level in the 1-month-old rats and was higher than in the single-group rats (Figure 6).

### 3.2. Morris Water Maze

*Total distance traveled*. Two-way ANOVA revealed a significant sex main effect on the average distance traveled on the first and third training days (*F*(1,65) = 8.399, *p* = 0.005 and *F*(1,65) = 14.157, *p* < 0.001, respectively) (Figure 7A). On the first training day, females traveled shorter distances than males, while on the third training day, their distance traveled was greater relative to males. There was no sex main effect for Probes 1 and 2, the Reminder trials, or Probe 3. The housing main effect and sex × housing interaction were not significant for distance traveled on either training days, the Reminder trials, or Probes 1–3.

*Average speed*. On the first training day, there were sex and housing main effects for the speed (*F*(1,65) = 11.469, *p* = 0.001, and *F*(1,65) = 15.900, *p* < 0.001, respectively). The sex × housing interaction was also significant (*F*(1,65) = 38.241, *p* < 0.001). The average speed of the single-housed females was lower relative to the other groups (Figure 7B). There was no difference on the subsequent training days, Probe 1, or Probe 2. The analysis of the Reminder trials yielded a significant housing main effect (*F*(1,65) = 6.326, *p* = 0.014). The average speed of the single-housed rats was lower compared with the group-housed rats. The sex main effect and sex × housing interaction were not significant. The analysis of Probe 3 revealed no differences.

*Latency to reach the area around the platform site*. Two-way ANOVA revealed a significant housing main effect on the index on the first training day (*F*(1,65) = 10.925, *p* = 0.002). The single-housed rats showed a longer latency to reach the area around the platform site (Figure 7C). There was no difference on training days 2–4, Probe 1, or Probe 2. For the Reminder trials, the analysis revealed a significant sex main effect (*F*(1,65) = 6.881, *p* = 0.011). Females needed more time to reach the area. There was no housing effect or sex × housing interaction. The analysis of Probe 3 revealed a significant housing effect (*F*(1,65) = 6.881, *p* = 0.011), but there was no sex main effect or sex × housing interaction. Latency to reach the area around the platform site was longer in the single-housed rats.

*Latency to reach the platform site*. On the first training day, there was a housing main effect for this indicator (two-way ANOVA, *F*(1,65) = 6.508, *p* = 0.013). The single-housed rats needed more time to locate the platform (Figure 7D). There was no difference on the second training day. On the third training day there was a sex main effect (*F*(1,65) = 7.781, *p* = 0.007). The latency to reach the platform site was longer in females. The analysis did not yield any difference between the groups for the fourth training day, Probes 1 and 2, the Reminder trials, or Probe 3.

*Relative time spent in the target quadrant*. There were no differences on training days 1 and 2. On the third training day, there was a significant sex × housing interaction (two-way ANOVA, *F*(1,65) = 5.210, *p* = 0.026). In females, the single-housed rats spent more relative time in the target quadrant. The single-housed and group-housed males did not differ (Figure 7E). The sex and housing main effects were not significant. The analysis did not yield any significant factor effects on training day 4. In Probe 1, one-sample Student’s t-test revealed that rats from all the groups spent significantly more relative time in the target quadrant comparing with the 25% random level (single-housed females: *t*(16) = 7.686, *p* < 0.001; single-housed males, *t*(16) = 6.303, *p* < 0.001; group-housed females, *t*(17) = 5.817, *p* < 0.001; group-housed males, *t*(20) = 7.701, *p* < 0.001). Overall, rats from all of the groups learned to locate the platform. Two-way ANOVA analysis of Probe 1 revealed a significant housing main effect (*F*(1,65) = 8.039, *p* = 0.006) and a significant sex × housing interaction (*F*(1,65) = 4.148, *p* = 0.046). The sex main effect was not significant. The single-housed females spent more relative time in the target quadrant compared with rats from other groups. When rats were 9 months old, there were no between-group differences for the probe trials and on the day of Reminder trials. However, in Probe 3, one-sample Student’s *t*-test revealed that only males spent significantly more relative time in the target quadrant comparing with the 25% random level (single-housed males, t(16) = 2.421, *p* = 0.029; group-housed males, *t*(20) = 2.233, *p* = 0.038; single-housed females, t(16) = 1.075, *p* = 0.299; group-housed females, *t*(17) = 1.620 *p* = 0.125). The data indicates that males show an advantage compared with females in spatial memory retrieval after the reminder.

*Relative time spent in the border area*. On the first training day, two-way ANOVA yielded a significant housing main effect (*F*(1,65) = 10.573, *p* = 0.001). The single-housed rats spent more relative time in the border area (Figure 7F). On training days 2 and 3, there were no differences between the groups. On the last training day, similar to the first training day, the housing main effect was significant (*F*(1,65) = 4.807, *p* = 0.032). Again, the single-housed rats spent more relative time in the border area. On training days, the sex factor showed no effect on relative time spent in the border area, and the sex × housing interaction was not significant. There was no difference in Probe 1, 2, and on the day of Reminder trials. The analysis of Probe 3 revealed a significant sex main effect (*F*(1,65) = 6.569, *p* = 0.013): The relative time spent in the border area was higher in females compared with males. The housing main effect and sex × housing interaction were not significant.

### 3.3. Passive Avoidance Paradigm

We excluded 10 out of 69 rats from the analysis due to the reasons described in Section 2.5. Hence, the following data represent the performance of 59 rats: 17 group-housed females, 14 single-housed females, 15 group-housed males, and 13 single-housed males.

Three-way repeated measures ANOVA revealed significant housing (*F*(1,55) = 29.360, *p* < 0.001) and stage (*F*(3,165) = 16.963, *p* < 0.001) main effects on the latency to enter the dark compartment and a significant housing × stage interaction. The sex main effect, sex × housing interaction, and sex × stage interaction were not significant. The performance of the rats in the PA paradigm depended on the housing conditions: The rats that lived in the social isolation failed to develop a conditioned PA (Figure 8). Post hoc analysis revealed that the initial values of the latency to enter the dark compartment of the arena measured at the Habituation stage did not differ between the groups. After the acquisition, at Retention 1 and Retention 2, the group-housed rats showed longer latency to enter the dark compartment relative to Habituation. In the single-housed rats, the latency to enter the dark compartment did not change throughout the experiment. At Retention 1 and Retention 2, the latency to enter the dark compartment was longer in the group-housed rats compared with the single-housed rats. At Retention 1, the percentage of the single-housed females that did not enter the dark compartment was lower relative to the group-housed females (0% v.s. 35.3%, Fisher’s exact test, *p* = 0.021). The percentage of the single-housed males that did not enter the dark compartment was also lower relative to the group-housed males (0% v.s. 33.3%, Fisher’s exact test, *p* = 0.044). There was no such difference for Retention 2.

At Retention 1, the single-housed females demonstrated fewer peeps into the dark compartment than group-housed females (0.4 ± 0.2 v.s. 3.3 ± 0.9, Mann–Whitney U test with FDR control, *p* = 0.003,). The quality (in points) of the PA conditioning did not differ between groups. At Retention 2, the number of incomplete transfers from the light to the dark compartment in the single-housed females was lower than in the group-housed (0.4 ± 0.2 v.s. 1.9 ± 0.6, *p* = 0.008). The quality of the conditioned PA reflex decreased in the single-housed rats compared with the group-housed animals. In the single-housed versus the group-housed animals, reflex points were: in males, 1.0 ± 0.0 and 1.9 ± 0.3 (*p* = 0.022) and in females, 1.1 ± 0.1 and 1.8 ± 0.2 (*p* = 0.010).

### 3.4. Body Weight, Serum Corticosterone Concentrations and Thymus, Spleen, and Adrenal Weights

Three-way ANOVA revealed significant sex (*F*(1,195) = 182.824, *p* < 0.001) and age (*F*(3,195) = 2262.234, *p* < 0.001) main effects on body weight. There was age × sex interaction (*F*(3,195) = 1792.154, *p* < 0.001). With age, body weight increased in male and female rats; females weighed less than males, starting at three months of age. There were no housing main effect (*F*(3,195) = 0.860, *p* = 0.357), sex × housing (*F*(3,195) = 0.428, *p* = 0.515), age × housing (*F*(3,195) = 2.133, *p* = 0.097), and sex × age × housing (*F*(3,195) = 0.888, *p* = 0.449) interactions. Weight changes occurred regardless of the housing conditions. However, at sacrifice, two-way ANOVA revealed not only significant sex (*F*(1,64) = 381.337, *p* < 0.001), but also housing (*F*(1,64) = 4.568, *p* = 0.036) main effects on body weight. The sex × housing interaction was still not significant. Females weighed less than males, and single-housed rats weighed less than group-housed rats. Because there was a sex main effect but no sex × housing interaction, we examined the housing main effect separately for males and females (Table 1). The analysis yielded no differences between the groups for the thymus relative weight. There were significant sex (*F*(1,64) = 10.208, *p* = 0.002) and housing (*F*(1,64) = 8.286, *p* = 0.005) main effects for the spleen relative weight; there was no sex × housing interaction. Females had a higher relative spleen weight than males, and single-housed rats had less relative spleen weight than group-housed rats. ANOVA revealed significant sex (*F*(1,64) = 294.042, *p* < 0.001) and housing (*F*(1,64) = 7.105, *p* = 0.009) main effects on relative adrenal weight and a sex × housing interaction (*F*(1,64) = 9.370, *p* = 0.003). Females had greater relative adrenal weight than males. Social isolation increased relative adrenal weight in females.

Two-way ANOVA revealed a significant sex main effect (*F*(1,36) = 25.565, *p* < 0.001) on blood serum CORT concentration: Females had higher CORT concentrations than males. There was not a housing effect or sex × housing interaction. The isolation affected the basal CORT levels neither in males nor in females.

## 4. Discussion

To the best of our knowledge, this study is the first to analyze the consequences of long-term social isolation from weaning up to the age of 10 months in males and females at the same time and using a wide range of behavioral indexes.

### 4.1. Open Field

We observed the dynamics of locomotor activity in socially isolated Wistar rats of both sexes compared with locomotor activity of control animals in two OF tests with different levels of environment-induced stress—aOF and cOF. Socially isolated rats showed essentially different results in these tests. We failed to detect any housing effect in the aOF test under relatively low environment stress—in a small arena under mild room light. Nevertheless, in the cOF test—in a large white arena under bright light—the housing effect was statistically significant: The locomotor activity of socially isolated Wistar rats was increased regardless of sex.

In socially isolated rats, increased sensitivity to new environments depends on aversive properties of the environments in which the behavioral assessment occurs [43,48]. These properties include the light intensity in the OF test. Researchers frequently fail to find changes in locomotor activity in socially isolated rats of various lineages in an unfamiliar small arena under mild white light. Thus, after 5 weeks of post-weaning social isolation, male Sprague Dawley rats increased locomotor activity in the OF under a low light level (25 W red bulb) [49], and after 8 weeks of post-weaning social isolation, there were no differences in locomotor activity under low light level (7.5 W red light) [50]. After 12-week social isolation, locomotor activity of male and female Sprague Dawley rats in a 60-min OF test with mild stress levels (12 lx) also did not differ from the activity of group-housed rats [32]. The same study reported motor hyperactivity in male Lister Hooded and Wistar rats after 3 months of social isolation. Note that the lack of motor hyperactivity in socially isolated Sprague Dawley rats is often denoted as a feature of the lineage [51]. Such a feature is not attributed to Lister Hooded and Wistar rats, yet social isolation also did not always lead to motor hyperactivity in these rat lines. Thus, 4 weeks of social isolation increased locomotor activity in male Lister Hooded rats [52], and after 9 weeks of social isolation, an analysis of the time spent in motion did not reveal any increase in a 10-min OF test with mild stress level (20 lx) [53]. Another study also did not reveal increased locomotor activity in male Lister rats after 12 weeks of social isolation in mild stress OF test [54]. Karim and Arslan [55] reported that the increase in locomotor activity in male Wistar rats after 6 weeks of social isolation did not reach statistical significance. We previously employed 3- and 10-min aOF (42 lx) tests and observed no difference in locomotor activity between group-housed male Wistar rats and rats reared in social isolation for 1 and 2 months starting from weaning [41,42]. However, when we prolonged social isolation up to 3 months (rats were 4 months old), locomotor activity in socially isolated rats was higher than in control rats—assessed by overall distance traveled in a 10-min aOF test [42]. These results are consistent with the data showing locomotor hyperactivity in single-housed Wistar rats as judged by the first 10 min of OF testing under low-stress conditions after 3 months of social isolation [56,57]. However, in our study, we decided not to interpret the data as locomotor hyperactivity in socially isolated male rats. This conclusion followed the analysis of locomotor activity changes with age in rats housed singly or in groups. We revealed that only in the 4-month-old group-housed males was locomotor activity decreased compared with baseline values measured when the rats were 1 month old [42]. In the current longitudinal study, we also found a decrease in locomotor activity in the OF test compared with baseline values in 5- and 8.5-month-old group-housed male Wistar rats. In the single-housed male rats, there was also a decrease in locomotor activity, but it did not reach statistical significance (Figure 2B); the locomotor activity of socially isolated rats did not differ from that of the group-housed rats at any observed age. These data are inconsistent with the findings reported by other authors concerning locomotor hyperactivity in male Wistar rats after 8 [51] or 12 weeks of social isolation [56,57]. In principle, a decrease in locomotor activity with age in the group-housed rats by itself may lead to differences between group-housed and single-housed rats. To summarize, we believe that the age-related dynamics of locomotion in Wistar rats in the aOF test question the conclusion about locomotor hyperactivity in socially isolated animals. In our opinion, to make a definitive conclusion on the presence or absence of changes based on this index in the socially isolated rats, it is essential to compare single-housed and group-housed rats at different ages to account for age-related alterations in locomotor activity. 

Another possible explanation for why we did not observe locomotor hyperactivity in male Wistar rats is different weaning ages. In other studies [51,53], pups were weaned from the dams at PND21, while we had weaned pups at PND24 and PND26 [41,42]. PND21, 24, and 26 are classified as the same developmental stage, either pre-adolescent, PND21–28 [58], or early adolescent, PND21–34 [13]. However, there are day-to-day differences in neurochemical and neurophysiological processes. A continuous increase in the dopaminergic system occurs in the CNS, accompanied by the completement of cortical synaptogenesis (for a review, see [59]). Given the data cited in that review, we suggest that interference with these processes at different days of the pre-adolescent stage, for example, by separating pups from the dams, may have a different effect on behavior, including locomotor activity. This assumption requires additional experimental verification.

In females, the age-related dynamics of locomotor activity essentially differed from that of male rats (Figure 2A). At 3 months old, females’ locomotor activity exceeded the initial level assessed at 1 month. The increase in locomotor activity was observed throughout the experiment (up to 8.5 months of age). The group-housed females most likely made the main contribution to the increase in locomotor activity since age-related changes in locomotor activity of single-housed females did not reach statistical significance (Figure 2B). Our data is generally consistent with the results of other researchers, who have reported higher locomotor activity in females relative to males in a low-stress arena in a range of rat lineages: Sprague Dawley, Long–Evans, Wistar, MR and MNR, and Albino [32,60,61]. Nevertheless, other studies did not register such an effect [60,62]. The higher locomotor activity of female rats is consistent with the notion that in females, aversive stimuli cause less pronounced behavioral inhibition than in males [63]. The sex-dependent differences in responses to stress are possibly due to hormonal actions [63] as well as a difference in chronology of monoaminergic pathways formation in male versus female postnatal brain [58]. 

Isolation is known to affect the estrous cycle in female rats: The incidence of acyclic state termed constant estrus was nearly twice as high in females living in isolation than in groups [64]. Locomotor activity increases during estrous (for a review, see [65]). However, in our study, single-housed females’ motor activity did not significantly differ from group-housed females’ motor activity throughout the entire experiment. Special studies are needed to identify the hormonal contribution to the increase in females’ motor activity during long-term social isolation. 

Of interest, aging may reduce locomotor and exploratory activity in rats [66], and itself act as a stressor occluding the influence of new-onset stressors [67]. At the end of our experiment, the rats were at 10 months of age, and they should be regarded as adults, but not yet old animals. However, we cannot rule out the impact of social isolation on the brain’s aging processes. Given this, the analysis of age × housing interaction deserves special attention. As shown in the “Results” section, we found no statistically significant age × housing interaction when analyzing locomotion in OF. This fact gives reason to believe that isolation determines those small differences that we identified in the locomotor activity in isolated rats versus group-housed animals.

Summarizing the current data, the locomotor activity of single-housed and group-housed Wistar rats of the corresponding sex changed in the same direction with age. However, in the single-housed rats, the changes were more subtle. The reason remains to be seen.

Recent studies with Wistar rats in cOF under bright light (350 lx) have revealed age-related differences in locomotor activity between naïve male and female rats: After 3 months, males decreased their locomotor activity, while females increased it, which led to sex differences in this index [46]. In our study, the locomotor activity of both female and male rats in the cOF decreased with age even under brighter light (approximately 500 lx), but the locomotor activity remained higher in females throughout the experiment (Figure 5). On the one hand, this finding is consistent with the notion that in females, behavioral inhibition is less pronounced in an aversive situation. On the other hand, it focuses on the dependence of the locomotion level on the situation stress level. Rat locomotor activity is lower in an OF performed under bright compared with soft light [68]; this finding is consistent with our data, especially in females. Nevertheless, the stress-inducing OF test has its benefits in resolving some issues. In the stress-inducing cOF test, but not in the aOF test, we observed a social isolation effect on rat locomotor activity: It was higher in the single-housed rats regardless of sex. However, there was no difference between the dynamics of locomotor activity of male and female single-housed rats, results that are the same as in the aOF test.

We thus decided to conduct an additional analysis of the habituation pattern in males and females in aOF. Sex and not the housing condition determined the rate of short-term habituation within the experimental session. Both males and females exhibited habituation, but in the 3- and 8.5-month-old females, the locomotor activity decreased slower than in males (Figure 3), a phenomenon that is probably associated with their higher stress reactivity. In a previous study that employed a low-stress-inducing aOF, with a similar assessment of the short-term habituation, the authors did not emphasize the differences in the habitation rate between males and females within the first 5 min of the session [69]. Nevertheless, the reported data indicate possible sex differences in this index. Given that a higher habituation rate reflects more effective information processing [70], we suppose that female rats are less effective in information processing than male rats in the OF conditions. More research is needed to test this assumption.

In the current study, we did not find any habituation impairment traits in the aOF for males or females after 2, 4, and 7.5 months of social isolation. Hence, we suggest that long-term social isolation leads to adaptive changes compensating for the single-housing condition, which neutralizes the negative influence of social isolation. With regard to males, the absence of habituation impairment in the OF has been reported by other authors in Wistar rats after 12 weeks of social isolation [56,57]. In our previous studies, we also did not find any disruption of habituation assessed by locomotor activity in the OF in male Wistar rats after 8 weeks of social isolation, but 12-week social isolation did induce habituation deficiency [42]. The fact that in the present study we did not observe such changes in the single-housed males suggests that locomotor habituation impairment is not one of the signature traits commonly induced by social isolation.

In the low-stress OF in adult females, the increased locomotor activity accompanied a sustained decrease in vertical exploratory behavior. In males compared with females, we consistently recorded lower locomotor activity and low levels of exploratory behavior. Even though vertical activity was decreased in the rats of both sexes, it was higher in females than in males (Figure 4). At first glance, these data on locomotor activity and exploratory behavior contradict other researchers’ reports showing that locomotor hyperactivity of socially isolated rats in the low-stress OF associated with enhanced exploration [49,52]. However, we did not observe the locomotor hyperactivity in aOF in males or females in the current work; there were only between-sex differences in locomotion. In our work, the absence of the enhanced exploration in the aOF test in the single-housed Wistar rats may indirectly evidence the lack of motor hyperactivity. However, in the high-stress cOF test, we detected signs of locomotor hyperactivity in the single-housed rats. Nevertheless, instead of the expected increase in vertical activity, we only observed an age-related decrease of exploration in both male and female rats. The alterations in vertical and horizontal activity presumably depend on the procedure stress level and may occur independently from one another. For example, the data reported by Varty et al. [50] support this assumption. In that article, the authors found an increase in vertical activity in the single-housed Sprague Dawley male rats after 8 weeks of social isolation, but there was no motor hyperactivity in a low-stress OF test.

Reactivity to novelty in rats implies an elevation of specific exploratory behavior in an unfamiliar environment or response to a new object or stimulus, which is associated with the detection of changes in the surroundings and stress-response activation [71]. Even though socially isolated rats usually react to a new environment by motor hyperactivity, additional behavioral analysis suggests the isolated rats exhibit deficits in reactivity to novelty. Gentsch et al. [72,73] assessed reactivity to novelty in the single-housed rats in an unfamiliar aOF by a reduction in the number of fecal boli, a decrease in reaction to post-habituation modifications of the OF, and a decline in the serum CORT level after a 10-min OF test. The authors suggested that the isolation-induced locomotor hyperactivity in the open field represents unfocused activity that does not facilitate but rather inhibits the animal’s proper recognition of the surroundings. Therefore, it is not sufficient to assess reactivity to novelty only by locomotor activity in the OF environment. For a more detailed analysis of behavior impairments in rats due to their prolonged social isolation, we tested reactivity to novelty using a simple and well-established method of rapid illumination change in the cOF test. After ≥ 4 months of social isolation, the rats of both sexes showed a deficit in reactivity to novelty (Figure 6), which is consistent with the previously obtained data on impaired reactivity to novelty in rats even after short-term social isolation [72,73,74]. There may be a direct link between motor hyperactivity and hyporeactivity to novelty in socially isolated rats. Unfortunately, we did not assess reactivity to novelty in response to a new stimulus in the aOF or by placing the rats in an unfamiliar environment different from OF.

### 4.2. Morris Water Maze

Our study is the first to report a new approach to the MWM. To an original experimental procedure, which includes a memory test 24 h from the last training trial (Probe 1), we added a memory test 4 months from the first memory test (Probe 2), two Reminder trials on day after Probe 2, and a third memory test 24 h from the last Reminder trial (Probe 3). This approach allowed us to gain more information concerning the efficiency of spatial memory formation and storage in the single-housed rats. There were statistically significant differences between the single-housed and the group-housed rats in the Reminder trials in average speed (the single-housed rats moved slower) and in the following Probe 3, in the latency to reach the area around the platform site (the single-housed rats reached this area later) (Figure 7). In Probes 1 and 2, we failed to detect any differences in these indexes. The lower speed of the single-housed rats in the Reminder trials and their higher latencies to reach the area around the platform site in Probe 3 did not affect the latency to reach the platform at the stated experimental stages. What possible explanations can there be? In a different Lister Hooded rat study, Lapiz et al. [75] suggested that the lower speed of individually housed (for 4 weeks) rats in the probe after the skill acquisition (which corresponds to Probe 1 in our design) may reflect a motivational deficit, while more platform area crossings may indicate better memory retention. We only observed speed reduction in rats from both sexes after 8.5 months of social isolation in the Reminder trials but not in Probe 3, so we do not attribute this finding to a motivational impairment in the single-housed rats. As stated above, higher latencies to reach the area around the platform site in the single-housed rats registered in Probe 3 after the Reminder trials did not result in an increase in the latency to reach the platform area (see Section 3.2) or in a decrease of the platform area crossing number (housing effect on crossing the platform site was not significant; *F*(1,65) = 0.715, *p* = 0.401), and there was no sex effect or sex × housing interaction (*F*(1,65) = 0.769, *p* = 0.384). Hence, we cannot say that the single-housed rats performed better or worse in this memory retention test. The group-housed rats, when moving faster and reaching the area around the platform site sooner, possibly swerved more often, changing the direction, and thus did not get to the platform site. By contrast, the single-housed rats reached the area around the platform site later but more often continued their way to the site where the platform was previously located. If we accept this explanation, we may assume that social isolation affects spatial memory retrieval: On the one hand, it slowed retrieval down, while on the other hand, recalled information allowed the rat to locate the platform more precisely. Whatever the answer, it must be recognized that neurophysiological processes underlying the single-housed rats’ behavior require a particular investigation. 

There were notable differences between the single-housed and the group-housed rats on the first training day: The single-housed rats reached the area around the platform site and the platform site itself at a later time. One possible explanation is the fact that on the first training day, the single-housed rats spent more time in the border area of the pool (Figure 7). We registered a similar effect in the MWM in the single-housed male Wistar rats after 3 months of social isolation [42]. We attribute this behavior from the isolated rats on the first training day to a greater thigmotaxis in a new stressful environment of the MWM, similar to the increase in thigmotaxis reported in the single-housed Long–Evans rats (26 days of social isolation) in an unfamiliar OF [76]. Enhanced thigmotaxis when a rodent is first placed in the MWM is considered a primary response of a highly emotional animal to a stressful situation [77]. We also recorded an increase in thigmotaxis on the fourth training day. We may assume that the activity of brain structures involved in early stress response is changed in the single-housed rats. Of note, 4 months after the training, we did not see any increase in thigmotaxis in the single-housed rats in Probe 2, Reminder trials, and Probe 3. Should this finding be considered an age-related decrease in reactivity to a stressful situation in the single-housed rats or their adaptation to aversive factors? Another explanation could be that the absence of enhanced thigmotaxis when rats were placed in the MWM after 8.5 months of social isolation might implicate situational memory retention without spatial memory retention.

Learning processes revealed sex differences in several indicators. On the first training day, females had a lower overall distance traveled compared with males. This phenomenon could be explained by the slower speed of the females. Indeed, there was a speed reduction, but only in single-housed females (Figure 7A,B). This group probably contributed most to the increased time spent in the border area and, as we believe, was more receptive to the stress induced by a new aversive environment. On the contrary, on the third training day, females’ distance traveled exceeded that of males; on that day, females had longer latencies to reach the platform. That was also the day when the single-housed females spent more relative time in the target quadrant than the group-housed females. Among others, the relative time spent in the target quadrant is commonly used as one of the indicators of learning efficiency in the MWM [78]. In Probe 1, rats from all groups preferred the target quadrant (Figure 7E), data that demonstrate long-term reference memory acquisition. However, the single-housed females spent the longest time in the target quadrant. Could we say that learning was most effective in the single-housed rats when there were no differences in the latency to reach the platform site and the area around it? Our approach, including an additional memory retention assessment 4 months after the training, reminder trials, and one more memory testing, provided supplementary data on sex differences in the processes of memory acquisition, retention, and retrieval in the rat. In the Reminder trials, females reached the area around the platform site at a later time compared with males, and after the reminder, in Probe 3, they spent more time in the border area (Figure 7C,F). In both these cases, the sex × housing interaction did not reach the accepted significance level, but the trend was evident (*F*(1,65) = 1.833, *p* = 0.181 and *F*(1,65) = 3.335, *p* = 0.072, respectively). The single-housed females likely most contributed to the longer latency to reach the area around the platform site in the Reminder trials and increasing the time spent in the border area in Probe 3. In support of this assumption are the data that in Probe 3, a later latency to reach the area around the platform site in isolated rats could have also been due to the single-housed females, because the sex × housing interaction almost reached the accepted significance level (*F*(1,65) = 3.969, *p* = 0.0506). Considering all this information, we believe that a spatial learning and memory deficit in females subjected to long-term social isolation may result from their greater vulnerability to the stress of exposure to the aversive situation of the MWM.

Spatial memory and navigation impaired with age [79]. Hence, we cannot ignore the natural aging impact on spatial memory, primarily in female rats. No aged female rats were observed estrous cycling regularly [80], but memory decline is associated with the loss of estrous cycling [79]. The above emphasizes the need to consider the estrous cycle’s influence on learning and memory under long-term social isolation.

In the current study, we did not find a clear advantage of male compared with female rats in spatial learning and memory retention (except for less time spent in the border area in males in Probe 3). At first glance, our data do not support other researchers’ conclusions that male rats learn faster and perform better in the MWM than females [81,82,83]. However, those conclusions were made concerning Sprague Dawley, Long–Evans, and Fisher rats. There was only a slight advantage (a tendency) in the MWM performance for male compared with female Wistar rats (for both group-housed and single-housed animals) [82]. We have come to a similar conclusion. In Probe 1, the relative time spent in the target quadrant did not reveal any differences in the learning efficiency and reference spatial memory. Nevertheless, Probe 3 revealed that only males, regardless of the housing condition, preferred the target quadrant. Thus, the applied approach with a reminder and additional skill assessment long after the training proved to be informative.

Retaining long-term memory for up to 4 months attracts attention. In a water maze of a different design using another algorithm, Martin et al. [84] found that sham-lesioned rats exhibited substantial forgetting of remote memory but still performed significantly better than chance in a retention test carried out 8 weeks after training. In our study, the memory retained for a period of twice as long. To obtain additional data on the spatial memory in 9.5-month-old rats, we had analyzed the change in the rats’ latency to reach the area around the platform in the MWM at two age points using the three-way ANOVA. We had chosen the first rat placing in the maze on the first training day (trial 1) when naive rats were 5.5-month-old and Probe 2 in 9.5-month-old rats. Based on the design, we could estimate only this index along the time axis. We assumed that the rats should forget the platform’s location and skill development situation four months after training. If the rats lose the skill, then the latency to reach the area around the platform should be the same as on the first trial. The analysis showed that there was only age main effect (*F*(1,44) = 32.921, *p* <0.001). There were no factor interactions. Four months after training, the rats reached the area around the platform faster than on the first trial at the first placing in the maze. These estimates also testified in favor of rats’ long-term spatial memory, albeit weakened. The effect did not depend on housing.

Fone and Porkess [85], in their review of spatial learning and memory in the MWM, reported an inconsistency in the accumulated data. We believe one of the reasons for such inconsistency (apart from rat strain, the time of onset and the duration of social isolation, and the training protocol) is the selection of indicators on which the authors base their conclusion on the efficiency of learning and memory (i.e., distance traveled, speed, latency to reach the platform site and the area around it, relative time spent in the target quadrant and border area, etc.) As the present study shows, selecting as few as two or three indicators is not enough. Obviously, more indicators complicate MWM behavioral analysis, and it may not be justified for some purposes. Nevertheless, this approach allows one to better understand a complex phenomenology of social isolation and finding new ways to identify the mechanisms underlying the recorded disturbances.

### 4.3. Passive Avoidance

The data obtained in the present study characterize, for the first time, the impairments of the associative learning in the rats of both sexes after 9 months of social isolation. Other researchers have shown that markedly shorter social isolation in different rat strains, namely, 8-week social isolation of Long–Evans rats [86] and 44-day [87] and 70-day [88] social isolation of Wistar rats, reduced PA. The current study results provide evidence that a long-term memory deficit in male rats in the PA paradigm is also present when social isolation is prolonged up to 9 months. Long-term memory impairments are also present in female rats. To the best of our knowledge, this study is the first to report a PA deficiency in female rats after long-term social isolation with an early-life onset. The latency to enter and the percentage of the animals that did not enter the dark compartment did not differ between the single-housed and group-housed males and females 24 h and 1 week after the acquisition. Thus, learning and memory in the PA paradigm were not sex dependent. This finding is consistent with the data reported by Ahmadian-Moghadam [89], obtained in 2-month-old naïve Wistar rats. 

Of note, Pavlova et al. [46] recently reported no sex differences in the transfer latency, but they revealed the following differences between naïve four-month-old male and female Wistar rats:PA reflex extinction within 7 successive days, starting from day 4, was slower in males than in females;In males, the reflex fear reaction was more profound than in females 24 h after the acquisition (assessed in qualitative behavioral scores).

In the present work, we did not use the reflex extinction paradigm. The data obtained 24 h after the acquisition showed no sex differences, which is consistent with the data reported by Pavlova et al. [46]. Summing up these findings and data obtained in our study, we suppose that in Wistar rats learning and memory in the PA paradigm are neither sex nor age dependent.

Along with the quantitative assessment of the PA reflex, we also applied the qualitative assessment proposed by Pavlova et al. [46]. The analysis revealed sex differences, but only between single-housed animals. In Retention 1, the single-housed females executed fewer peeps; in Retention 2, they made fewer incomplete transfers to the dark compartment than the group-housed females, an outcome that might have explained their lower PA score than that of the group-housed females. The single-housed males did not significantly differ from the group-housed males based on these indicators, but in Retention 2, their score was also lower than that of the group-housed males. In our opinion, the lower PA scores of the single-housed compared with the group-housed rats in Retention 2 provides evidence of less pronounced conditional reflex fear and lower emotional stress in the single-housed compared with the group-housed rats in the decision-making process to enter the dark compartment. However, the single-housed males showed greater conditioned reflex fear than the single-housed females. In this case, we can hardly associate the between-sexes differences with the estrous cycle impact in females: Direct evidence obtained that training with punishment in female rats does not fluctuate during the estrous cycle [90].

In a study from Inozemtsev [91], dedicated to the nature of associative processes in PA, the author concluded that increased passive response latency is not caused by the memory of the shock in the strictly certain place and accordingly not by shock avoidance in it, but by a nonspecific defensive response (freezing) that is unrelated to the shock location. If we accept this hypothesis, the following question remains: Why do socially isolated rats, showing decreased transfer latency to the dark compartment after the acquisition, have a lower defense reaction? In the same rats, we employed the “hot plate” test and showed that the pain thresholds in the single-housed rats were higher than in the group-housed rats after 2, 4, and 7.5 months of social isolation (in 3-, 5-, and 8.5-month-old rats, respectively); moreover, the jumping latency in the 5-month-old females was higher than in the same-aged males [92]. Lower pain sensitivity (hypoalgesia) caused by social isolation may contribute to lower defense reaction assessed by transfer latency in male and female single-housed rats. Nevertheless, some data oppose this assumption: A 3-day swimming test resulted in a decrease in pain thresholds (hyperalgesia) in the rats in the “hot plate” test, along with learning deficiency in the PA paradigm judged by lowering the latency to enter the dark compartment [93]. The interrelation of pain sensitivity and PA learning remains to be examined.

Locomotor activity changes could also decrease the transfer latency seen in the single-housed rats in the PA paradigm. As stated above, illumination in the light compartment was bright (156 lx). In the present study, we demonstrated that in cOF test under bright light, locomotor activity of the single-housed was higher than that of the group-housed rats. However, the fact that females showed greater locomotor activity contradicts this explanation. Indeed, if the locomotor activity determines the transfer latency, then we should have seen sex differences not only between the single-housed male and female rats but also between the group-housed males and females. However, we did not observe such differences.

Another explanation for decreased transfer latency to the dark compartment in the single-housed rats might be their anxiety level. Several studies have shown that social isolation increases anxiety in single-housed rats of different lines, including Wistar rats [94,95,96], but it is not always the case (for a review, see [85]). Horii [97] showed that PA was better in highly anxious rats, and Nazeri [93] reported that PA deficiency was accompanied by lowering the anxiety level. Hence, we suggest that in our study, the single-housed rats’ anxiety levels were lower compared with the group-housed rats (yet this supposition requires experimental testing). This assumption is also consistent with the suggestion given above that the single-housed rats have a lower nonspecific defense response and decreased reactivity to novelty revealed in the current study (see Section 4.1). Decreased reactivity to novelty is associated with memory deficiency in the novel object recognition task (a decrease in the discriminative index) [98]. We assume that reduced reactivity to novelty may be a prognostic sign of cognitive impairments.

### 4.4. Serum Corticosterone Concentrations and Thymus, Spleen, and Adrenal Weights

HPA axis activity in rodents is affected by sex hormone levels: In females, the baseline blood serum CORT level is higher than in males, and the stress-related CORT response is more pronounced in females [29,99]. In the current study, we also found a higher blood serum CORT level in females compared with males, although there were no differences between the group-housed and the single-housed rats of the same sex (Table 1). These data suggest that long-term aversive exposure, such as 9 months of social isolation with an early onset, did not impact the level of a principal glucocorticoid. Can we conclude that such prolonged social isolation is not a stress factor and does not change the functional state of the HPA axis? The scientific literature concerning the changes in rat basal blood CORT level (without additional acute stress) under early social isolation for considerably shorter periods includes controversial data. A reduction in the basal blood CORT concentration was reported after 8-week and 6-week social isolation in male Wistar rats [36,40]. Thirteen-day social isolation in male and female Sprague Dawley rats also caused a decline in the blood CORT level [29]. On the contrary, another study showed that 6-day social isolation in male Sprague Dawley rats increased the blood serum CORT level [38]. After approximately 14 days of social isolation in male Wistar rats [57] and 18 days of social isolation in male and female Sprague Dawley rats [32], kept under standard vivarium conditions, there were no changes in the basal CORT level. Thus, there is no clear association between CORT level changes and ratlines or duration of social isolation. Presumably, under social isolation, several factors impact the CORT secretion system, so only the presence of alterations in its reactivity and not the direction of changes in basal CORT level can be discussed. 

One way to assess the HPA axis response in rats subjected to long-term social isolation is to test their basal CORT level and its dynamics after an additional acute stress exposure [100]. However, this approach does not always allow one to determine the effect of social isolation [32]. In a previous study [40], social isolation stress decreased plasma CORT concentration, the subsequent acute restraint stress increased blood CORT levels in both group-housed and socially isolated rats. However, CORT concentration remained diminished in the single-housed rats when compared with the group-housed animals. Meanwhile, Pisu et al. [29] reported that an acute foot-shock caused a more pronounced increase in the single-housed rats’ CORT level and the effect was even more significant in males. The approach’s efficiency may depend on the kind of stress: Foot-shock stress is considered to be more severe because it relies on pain reinforcement. In the present study, we did not subject the rats to additional acute stress before taking blood samples. Nevertheless, we assessed the weights of stress-related organs, which directly regulate blood CORT concentration. 

Adrenal glands (the adrenal cortex) secrete CORT into the blood and regulate basal and functional hormonal levels [101]. The changes in their relative weight in socially isolated rats are sex-dependent: In male rats, there were no differences in adrenal gland weights (after 5, 6, and 8 weeks of social isolation in Wistar and Sprague Dawley rat lines) [36,37,38,39,40], while in females, the weights of the adrenal glands increased (13 and 14 weeks of social isolation in hooded and Wistar rats) [35,37]. Our results are in good agreement with these data. We should emphasize that our work employed much longer social isolation. Thus, the data on adrenal gland weights may provide evidence of relatively consistent sex differences in this endocrine organ state at different periods of exposure to post-weaning social isolation. It is well known that sex hormones regulate HPA axis activity and that male and female hormones have a different effect on the HPA axis state [25]. Despite the absence of differences in the CORT level between the group-housed and single-housed females, changes in the adrenal gland weights indicate alterations in the HPA state and, thus, chronic stress. Of note, CORT secretion may depend on the stage of the estrous cycle in females under stress. As shown in a previous study [102], restraint stress can even result in oppositely directed changes in CORT levels depending on the estrus cycle phase. Unadjusted data in females may be misleading.

In the current study, prolonged social isolation also reduced the spleen weight in male and female rats (only a trend in males), and body weight decreased only in males. There were no differences in thymus weights. Hatch [35] reported a decrease in the spleen, thymus, and body weights in male and female Wistar rats socially isolated for 13 weeks when describing “isolation syndrome.” The subsequent years have yielded research data in support as well as against this finding. For example, the weights of male Wistar rats subjected to 2 months of social isolation started on PND16 did not differ from control animals, and their thymus weights were even greater compared with the control group [40]. After 6 weeks of social isolation started at PND21, neither thymus nor body weights were affected [36]. Five-week social isolation in Sprague Dawley rats starting at PND24 decreased body weights and increased thymus weights in the rats [39]. Cruz et al. [103] found a reduction in male Wistar rats’ body weights after 3 weeks of social isolation started at PND28. In our previous studies, 2 and 3 months of social isolation started at PND24/26 did not alter body weights of male Wistar rats [41,42]. Changes in the thymus and body weights are presumably relatively unstable characteristics under social isolation stress.

It is known that in the lymphoid organs, CORT can be locally produced from its inactive metabolite, 11-dehydrocorticosterone (DHC), and vice versa, CORT can be converted to the non-active DHC [99]. Glucocorticoids, in turn, modulate lymphoid organs’ function. Specifically, CORT induces cell death in lymphoid organs, which is the major reason for their involution under stress; moreover, the severity of stress that impacts the lymphoid organs grows with the duration of stress [104]. Therefore, our finding on relative spleen weight decrease in rats after 9 months of social isolation indicates chronic stress in both male and female rats, but the effect is more pronounced in females. The lack of differences in thymus weights in males and females may be due to age-related involution of the organ [105], since the rats were 10 months old at the sacrifice. Females weighed less than males, especially at a later age, which is an expected and consistently reproduced result [105,106].

## 5. Conclusions

Prolonged social isolation, with onset at the edge of early-to-mid adolescence [13,58] and lasting up to 9 months, caused sex-dependent learning deficits in Wistar rats. In the low-stress OF test, we did not detect motor hyperactivity in male or female single-housed rats. Age-related changes in locomotor activity of the single-housed rats were in the same direction as that of the group-housed rats, but they were less pronounced. Habituation (non-associative learning) in 3- and 8.5-month-old females was slower. In the stressful OF test, there was motor hyperactivity in the single-housed rats regardless of sex as soon as after 2 months of social isolation. After 4 months of social isolation and throughout the experiment, the rats showed decreased reactivity to novelty, which was assessed by changes in locomotor activity in response to a sudden change in lightning. To examine learning and spatial memory, we employed the standard MWM scheme along with a new approach whereby we tested the skill three times: 24 h after the training, 4 months later before a reminder, and 24 h after the reminder. With regard to learning, the major differences between the single-housed and group-housed rats occurred on the first training day when the single-housed rats required more time to reach the area around the platform site and locate the platform while spending more time in the border area of the pool. In the Reminder trials, the single-housed rats moved slower, and 24 h after the reminder, they required more time to reach the area around the platform site than the group-housed rats. On the first training day, the single-housed females moved slower than the rats in the other groups. The single-housed and group-housed rats, regardless of sex, memorized the location of the platform—based on the preference of the target quadrant in the first probe trial—although, in the third probe trial after a reminder, only male rats regardless of the housing condition preferred the target quadrant. In the Reminder trials, females had higher latencies to reach the area around the platform site, and in the subsequent probe trial, they spent more time in the pool’s border area. Given the data on learning and spatial memory impairment in the single-housed rats and sex × housing interaction, we assume that the major contribution to the increased latencies to reach the area around the platform and increased time spent in the border area was due to the single-housed females. The deficit in spatial learning and memory in the single-housed females may be explained by their greater vulnerability to the stress of being in an aversive situation like the MWM. The performance in the PA task when quantitatively assessed did not depend on the sex of the animals but was disrupted in the single-housed rats. Additional PA scoring revealed that the quality of PA a week after the acquisition was worse in the single-housed compared with the group-housed rats. In addition, 24 h and 1 week after acquisition, the single-housed females executed fewer peeps and incomplete transfers to the dark compartment than males, thus showing less pronounced conditioned reflex fear memory. In both male and female rats, basal CORT levels in rat blood serum after 9 months of social isolation did not differ from that in the group-housed rats of the corresponding sex. Meanwhile, females’ basal corticosterone level was higher than in males, regardless of the housing conditions. Of note, the relative weight of the adrenal glands was only altered (increased) in the single-housed females. 

Long-term social isolation started early in life induces more pronounced cognitive impairments in the MWM and PA paradigm in adult single-housed females than males, perhaps due to greater female rats’ vulnerability to the impact of prolonged isolation.

## 6. Limitations

Our study has some limitations that we need to consider when evaluating results. We attribute to the design shortcomings a gap of 4 days on the social isolation onset in females and males, which was caused by the need to conduct primary behavioral assessments in a large number of animals for the further formation of groups that did not initially differ in significant indicators. We did not assess the estrous cycle stage’s impact on females, although it is essential for assessing behavior and stress vulnerability. In this work, we did not provide data on rats’ anxiety levels in the social isolation dynamics and aging; however, this data could help understand the results and discussion. We evaluated reactivity to novelty only by increasing locomotor activity when changing illumination in a classic OF and did not apply tests for recognizing a new object. The animals’ locomotor activity was assessed only in OF tests, although they differed in environment-induced stress. In the future, it is advisable to evaluate the age-related dynamics of locomotor activity in the group- and single-housed rats of both sexes in parallel in several tests that will help to more clearly separate the effects of aging and social isolation on animal mobility.

## Figures and Tables

**Figure 1 brainsci-10-00799-f001:**
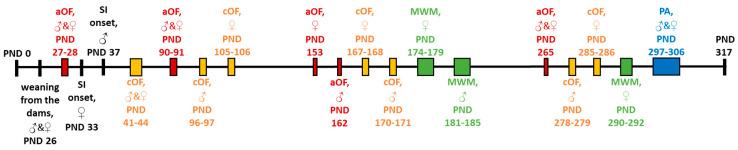
Age of behavioral assessment. Postnatal day 0 (PND0)–the day of birth; PND317–the day at which the animals were sacrificed; SI–social isolation; aOF–automated open field test; cOF–classic open field test; MWM–Morris water maze; PA–passive avoidance paradigm; ♀–female rats; ♂–male rats.

**Figure 2 brainsci-10-00799-f002:**
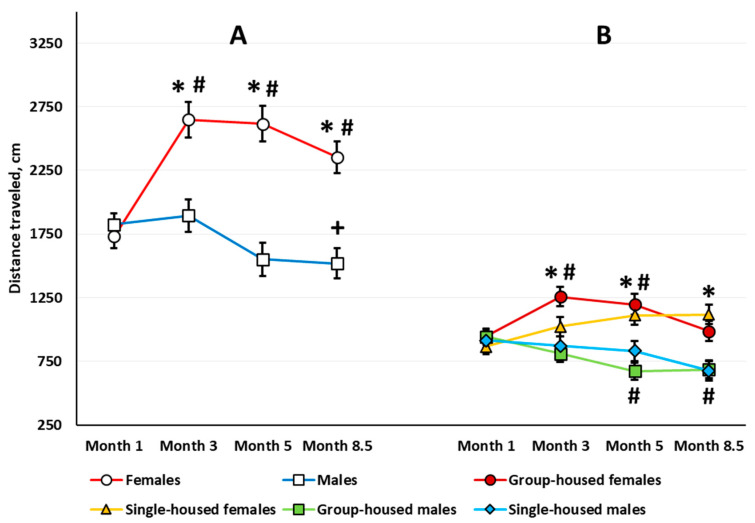
The total distance traveled (cm) for 10 min (sex main effect) (**A**) and the first 3 min (sex × housing interaction) (**B**) in the automated open field test. Here and in other figures, the horizontal axis indicates the age of the rats. The # indicates a statistically significant difference compared with the distance traveled in the 1-month-old rats of the same group, *p* ≤ 0.039; the * indicates *p* < 0.001 compared with the distance traveled in the males of the same age and group (single-housed or group-housed); the + indicates *p* = 0.044 compared with the distance traveled in the 3-month-old males (ANOVA followed by the Newman–Keuls post hoc test). See Appendix A.

**Figure 3 brainsci-10-00799-f003:**
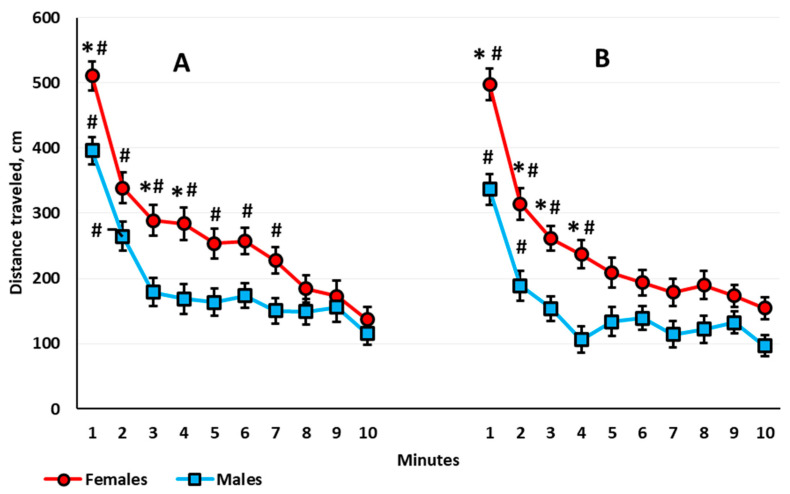
The distance traveled (cm) by female and male rats over 10 min in the automated open field test; (**A**) 3-month-old rats and (**B**) 8.5-month-old rats. The * indicates a statistically significant difference compared with the distance traveled in the males of the same age at the same minute, *p* ≤ 0.004; the # means *p* ≤ 0.015 compared with the distance traveled in the tenth minute by the rats of the same age and sex (ANOVA followed by the Newman–Keuls post hoc test). See Appendix A.

**Figure 4 brainsci-10-00799-f004:**
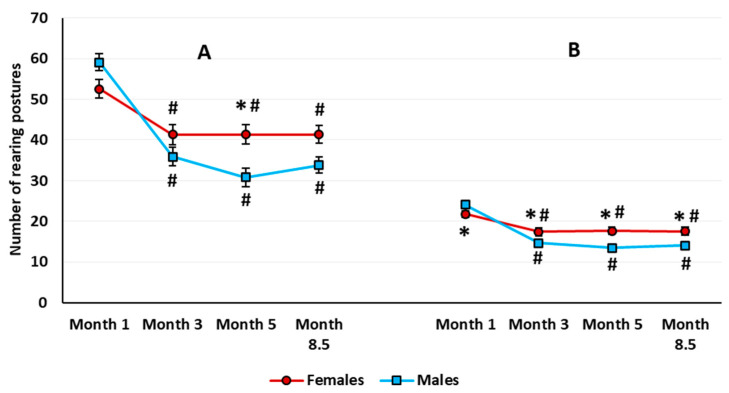
The total number of rearing postures over 10 min (**A**) and in the first 3 min (**B**) in female and male rats in the automated open field test. The # indicates a statistically significant difference compared with the number of rearing in the 1-month-old rats of the same sex, *p* < 0.001; the * indicates *p* < 0.001 compared with the number of rearing in males of the same age (ANOVA followed by the Newman–Keuls post hoc test). See Appendix A.

**Figure 5 brainsci-10-00799-f005:**
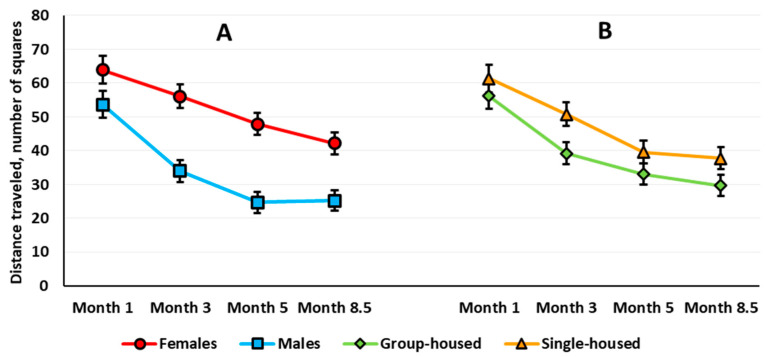
The total distance traveled (number of squares) for 3 min in rats in the classic open field test: (**A**) sex and (**B**) housing factors. See Appendix A.

**Figure 6 brainsci-10-00799-f006:**
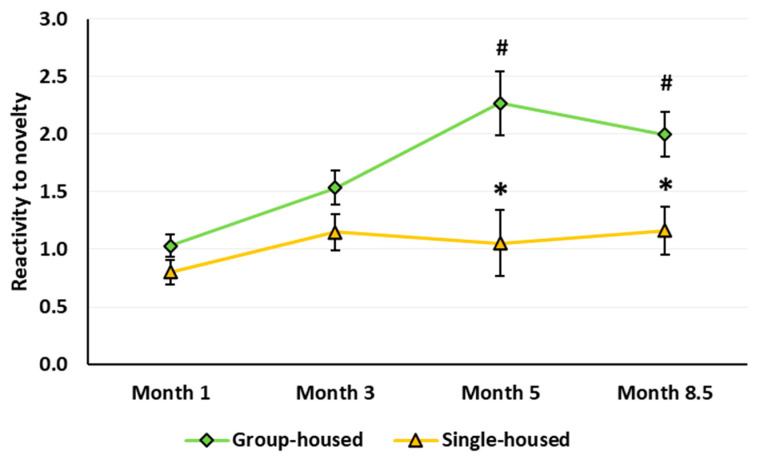
Reactivity to novelty (the ratio of the distance traveled in the first minute to the distance traveled in the fourth minute in the Classic open field test). The * indicates statistically significant difference compared with the group-housed rats of the same age, *p* < 0.007; the # means *p* ≤ 0.004 compared with the 1-month-old group-housed rats (ANOVA followed by Newman–Keuls post hoc test). See Appendix A.

**Figure 7 brainsci-10-00799-f007:**
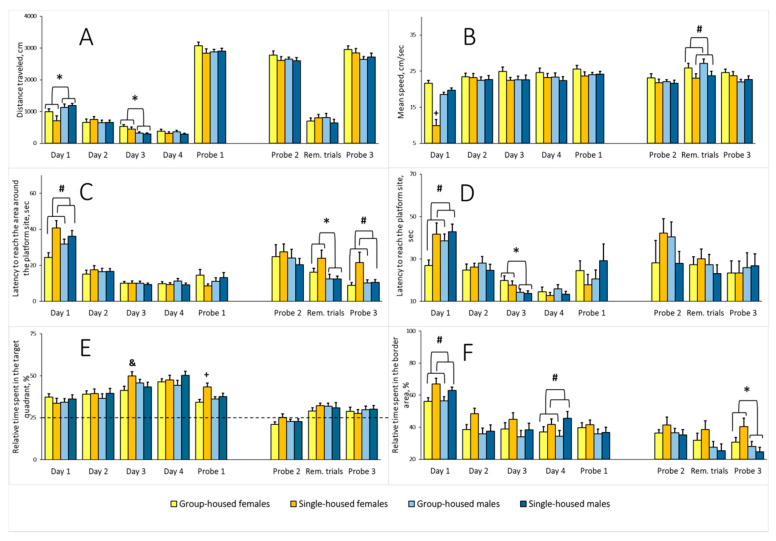
(**A**) Overall distance traveled in the Morris water maze (MWM), cm; (**B**) average speed in the MWM, cm/sec; (**C**) latency to reach the area around the platform site, sec; (**D**) latency to reach the platform site, sec; (**E**) relative time spent in the target quadrant, %; (**F**) relative time spent in the border area, %; 5.5-month-old rats: Day 1–4 and Probe 1; 9.5-month-old rats: Probe 2, the Reminder trials, and Probe 3. The * indicates a statistically significant difference for all females compared with all males, *p* ≤ 0.016; the # indicates *p* < 0.043 compared with all the group-housed rats; the + indicates *p* < 0.05 compared with all other groups (Newman–Keuls test). The & means *p* = 0.015 compared with the group-housed females (Fisher’s LSD test). Rem. trials–Reminder trials. See Appendix A.

**Figure 8 brainsci-10-00799-f008:**
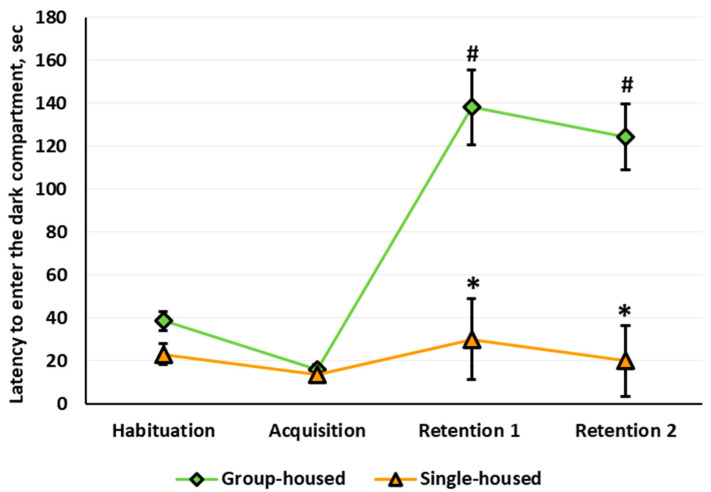
The latency to enter the dark compartment of the arena (sec) in the passive avoidance paradigm. The # indicates a statistically significant difference compared with the group-housed rats at the Habituation stage, *p* < 0.001; the * indicates *p* < 0.001 compared with the group-housed rats at the same stage of the experiment (ANOVA followed by the Newman–Keuls post hoc test). See Appendix A.

**Table 1 brainsci-10-00799-t001:** Body weight (g), relative weights of thymus, spleen, and adrenal glands (mg/g BW), and serum corticosterone concentration (ng/mL) in control and socially isolated rats.

	Group-Housed Females	Single-Housed Females	Group-Housed Males	Single-Housed Males
**Sample size (*N*)**	17	16	20	15
**Body weight**	308.00 ± 9.59 *	296.93 ± 9.89 *	505.90 ± 8.84	475.7 ± 10.21 #
**Thymus**	0.92 ± 0.05	0.77 ± 0.05	0.75 ± 0.06	0.75 ± 0.05
**Spleen**	3.51 ± 0.19 *	2.90 ± 0.19 #	2.84 ± 0.17	2.37 ± 0.20 +
**Adrenal glands**	0.20 ± 0.01 *	0.24 ± 0.01 * #	0.11 ± 0.01	0.10 ± 0.01
**Corticosterone**	34.48 ± 3.56 *	40.58 ± 5.13 *	17.92 ± 2.89	17.71± 2.31

* Indicates a statistically significant difference (*p* < 0.001) compared with male rats of the same group (single-housed or group-housed); # means *p* < 0.031; + indicates a trend (*p* = 0.082) compared with group-housed rats of the same sex (two-way ANOVA followed by the Newman–Keuls post hoc test).

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
