# Peer review of "Prolonged Social Isolation, Started Early in Life, Impairs Cognitive Abilities in Rats Depending on Sex"

_brainsci, 2020, doi:10.3390/brainsci10110799_

Round 1
Reviewer 1 Report
In this paper, the authors evaluated the impact of early, long-term social isolation on various behavioral functions including cognitive functions, reactivity to novelty and overall locomotor activity. They included both males and females, which is appreciated. Rats are socially isolated from 1 month of age until 9-10 month of age. Behaviors are tested at different time-points during this experimental timeframe. The experiments and data analyses were conducted with rigor. However, the lack of any brain analyses that could explain the results or provide some insights into the mechanisms by which social isolation impact behaviors is puzzling. Also, some aspects of the methods need to be more detailed.
Why is one month of age different from males and females?
For the MWM, I would not expect any rat to remember a target quadrant exposed to 4 months earlier. Is there any literature that support the idea that aging rats (9-month-old!) have long-term memory that can last for 4 months?
Under which conditions blood and tissue collection was performed? Since the authors measure CORT it is important to describe how and when was the collection performed as experimental conditions can impact the results.
The authors failed to include discussion about the potential contribution of normal aging processes to the behavioral results. When testing rats at 10 months of age we can anticipate an impact of aging on brain functions. Normal aging processes are likely to be affected by a life-long social isolation.
The authors should discuss the potential contribution of estrus cycle, and estrus cycle x age, in females. Many of the behaviors reported here have been shown by others to be impacted by the estrus cycle. Some discussion on this topic should be included.
The statement “prolonged social isolation with an early adolescent onset are associated with the early-life reorganization of the neural circuits mediating cognitive functions” is an over-statement since no analysis of the brain was conducted. Some of the effects observed might be driven by the effect of social isolation on aging processes.
Reviewer 2 Report
The authors of the manuscript “Prolonged Social Isolation, Started Early in Life, Impairs Cognitive Abilities in Rats Depending on Sex” evaluated sex differences in cognitive function following protracted social isolation in Wistar rats in attempt to address inconsistencies in the literature. Here, they report sex-dependent cognitive impairments in OF habituation and initial MWM training, with outcomes worse for females, and greater conditioned fear in males. Interestingly, while report in the Abstract that basal CORT did not differ between sexes, female increases were found in adrenal glands; in the body of the text (p. 13), however, they do indicate basal differences between females and males (Table 1). The work is laudable in its design and the multiplicity of behavioral assessments made alongside postmortem organ weight assessments. Some of the presentation of the information is confusing and controversial (i.e., from one section to another), and the Discussion is entirely too lengthy in its current form.
The authors need to reconcile:
Differences reported in the Abstract and Results section (i.e., differences in organ weight and basal CORT). As indicated above, in the Abstract they report no differences in basal CORT but at the end of the Results, they indicate females’ basal CORT levels are higher. Finally, in the Conclusions, the authors report that there are no differences in basal CORT. The data in Table 1 and the Results suggest otherwise. Which is it?
Not asserting that they are looking at 10-month isolation stress when assessments are taking at 3, 6, 9 months for OF analyses.
The authors report complex significant interactions for the OF tests (i.e., sex x housing x age) and yet pictorially flatten this interaction by only presenting the data by sex or by group-housing. I’m wondering if there is a better way to demonstrate the nuances of the interactions that they observe in OFA (c & a) – they do so for the MWM data. It would be interesting, for instance, for them to display the sex and housing effects, perhaps by plotting males and females side by side for each: housing conditions and age?
Another interesting question the authors should address is how they might distinguish simple age-related changes with changes as a function of time in isolation? It seems to be a confound since they did not assess locomotion in any other task. Similarly, what about the weight disparities in the group- versus single-housed rats, and in males versus females across the age axis. Could either of these variables also contributed to variations in mobility? How would the authors discern if, indeed, it is “learning” that impacts how mobile the rats are as they age and gain weight, and/or as they have time in select housing conditions?
The authors should truncate the Discussion, particularly the culminating Conclusions section. They are also encouraged to limit their discussion of reprogramming, reorganization, etc. as it is a stretch to relate it to their particular paradigms and model since there were not tested currently.
The authors might consider clearing describing their findings, followed by their interpretations and how they fit with previous work (in their hands, that of other labs). Next, they might write a separate Limitations section where they highlight what are the shortcomings of the design, the cautions with interpretation, etc.
Round 2
Reviewer 1 Report
The authors addressed most of previous comments. However, some were not correctly addressed.
- When asking under which conditions blood was collected, I didn’t mean the procedure itself, but rather, what happened to the animals in the minutes before blood sampling. Were they maintained in the colony room and directly decapitated? Where they stress (if so for how long and how long before sampling)? Where they brought into another room than the colony room and gave some time to recover?
- The authors did not address the lack of analyses conducted in the brain (why they were not done? What markers would they target in which brain regions if they were to do any molecular assay and that can support their findings/conclusions).
